# Cost-Effectiveness Analysis of Pitavastatin in Dyslipidemia: Vietnam Case

**DOI:** 10.3390/healthcare13192494

**Published:** 2025-10-01

**Authors:** Nam Xuan Vo, Hanh Thi My Nguyen, Nhat Manh Phan, Huong Lai Pham, Tan Trong Bui, Tien Thuy Bui

**Affiliations:** 1Faculty of Pharmacy, Ton Duc Thang University, Ho Chi Minh City 700000, Vietnam; h2000382@student.tdtu.edu.vn (H.T.M.N.); phanmanhnhat@tdtu.edu.vn (N.M.P.); h1900276@student.tdtu.edu.vn (H.L.P.); 2Faculty of Medicine, University of Medicine and Pharmacy at Ho Chi Minh City, Ho Chi Minh City 700000, Vietnam; bttan.y22@ump.edu.vn; 3Faculty of Pharmacy, Le Van Thinh Hospital, Ho Chi Minh City 700000, Vietnam; bttien.ths.tcqld23@ump.edu.vn

**Keywords:** cost-utility, cost-effectiveness, economic evaluation, pitavastatin, dyslipidemia, hypercholesterolemia, reduce LDL-C, quality of life, Vietnam

## Abstract

**Background/Objectives**: Dyslipidemia is becoming a significant economic healthcare burden in low- to middle-income countries (LMICs) due to its role in heightening cardiovascular-related mortality. Statins are the first-line treatment for reducing LDL-C levels, thereby minimizing direct costs associated with cardiovascular disease management, with pitavastatin being of the newest generation of statins. This research work conducted a cost-utility analysis of pitavastatin to determine the economic benefit in Vietnam. **Methods**: A decision tree model was developed to compare the rate of LDL-C controlled patients over a lifetime horizon among patients treated with pitavastatin, atorvastatin, and rosuvastatin. The primary outcome was the incremental cost-effectiveness ratio (ICER), measured from the healthcare system perspective. Effectiveness was evaluated in terms of quality-adjusted life years (QALYs), using an annual discount rate of 3%. A one-way sensitivity analysis was performed to identify the key input parameters that most influenced the ICER outcomes. **Results**: Pitavastatin was cost-effective compared to atorvastatin but was dominated by rosuvastatin. Although pitavastatin gained fewer QALYs than atorvastatin, the ICER was 195,403,312 VND/QALY, well below Vietnam’s 2024 willingness-to-pay. Drug cost had the most significant impact on ICERs. **Conclusions**: Pitavastatin represents an economical short-term alternative to atorvastatin, particularly in resource-constrained settings.

## 1. Introduction

Dyslipidemia is characterized by any abnormal blood lipid levels, including four main parameters: elevated total cholesterol (TC), triglycerides (TG), and low-density lipoprotein cholesterol (LDL-C), or persistently low high-density lipoprotein cholesterol (HDL-C) [1]. Among these abnormalities, high LDL-C is recognized as the most significant modifiable risk factor for the development of atherosclerotic plaques, leading to cardiovascular diseases (CVD) [2,3,4,5]. CVD remains a global health crisis due to its high mortality rates in both developed and developing nations, with ischemic heart disease being the leading contributor [6]. Approximately one-third of cardiovascular-related conditions are attributed to uncontrolled cholesterol levels [7]. Uncontrolled LDL-C is more concerning in aging populations, particularly among individuals aged 60–89, as they not only face a threefold higher risk of CVD-related death compared to younger adults, but also experience a significant decline in quality of life [8]. The burden is especially severe in low- and middle-income countries (LMICs), where CVD accounts for nearly 80% of deaths according to the World Health Organization (WHO) [6]. From 1990 to 2019, high-income countries saw a drop of 35% in stroke mortality caused by high LDL-C, whereas the rates stayed largely the same or even rose in Asian and African countries [9]. Such rising dyslipidemia prevalence has placed increasing pressure on healthcare systems, straining budgets due to the cost of managing heart-related complications and long-term medication use [10,11,12]. Dyslipidemia often progresses silently and is usually detected late through laboratory tests, which delays intervention and associates with more pressure on the healthcare system, along with increased premature mortality [13].

Reducing LDL-C to the optimal range is the prioritized goal to mitigate the risk of developing CVD and preserve treatment cost, with statins being the most efficient therapy as recommended by guidelines [8]. According to one review estimating the prevalence of hypercholesterolemia in Vietnam from 2010 to 2023, 67.6% of individuals were diagnosed with abnormal lipid profiles, highlighting the need for medical policies to select the most effective lipid-lowering medication to reduce the disease burden [14]. Pitavastatin, a newer-generation statin, is effective in lowering LDL-C even at low doses (1–4 mg/day) and has a favorable safety profile, with a reported rhabdomyolysis rate of only 0.01% [15,16]. It has a unique metabolic pathway through glucuronidation, rather than relying heavily on CYP450 enzymes like other statins, which minimizes drug–drug interactions [17]. This metabolic difference also influences hepatic LDL receptor activity, enhancing the lipid-lowering efficacy of pitavastatin even at low doses, making it comparable to high doses of commonly used statins such as atorvastatin or simvastatin [17,18]. Thus, pitavastatin appears to be a promising agent for hypercholesterolemia in the Asian context, particularly in Vietnam, where atorvastatin and rosuvastatin are the most commonly prescribed lipid-lowering agents. However, so far, there have been no economic evaluations to assess the cost-effectiveness of pitavastatin, which makes it difficult for stakeholders to justify its inclusion in national insurance coverage. Moreover, the high cost of pitavastatin 2–4 mg, VND 13,500–18,500 (0.56–0.77 USD), compared to atorvastatin, VND 15,941 (USD 0.66), and rosuvastatin, VND 8978 (USD 0.37), poses an access barrier for patients in Vietnam. This raises the question of whether its strong LDL-C lowering effect justifies the high acquisition cost in hospital settings, and at what budget threshold pitavastatin could improve quality of life. This study aims to assess the cost-effectiveness of pitavastatin compared to atorvastatin and rosuvastatin in the treatment of dyslipidemia in Vietnam, based on the capability to achieve target LDL-C levels. The findings aim to inform health policy recommendations, such as including pitavastatin in the national insurance reimbursement list or implementing subsidy programs to optimize healthcare resource allocation and improve access to lipid-lowering therapy.

## 2. Materials and Methods

### 2.1. Study Design and Patient Population

This study employed a cost-effectiveness analysis (CEA) approach to evaluate the economic effectiveness of pitavastatin in treating dyslipidemia.

The study population was sourced from an economic analysis conducted in South Korea in 2017 by Jeong et al. [19]. The initial patient cohort consisted of patients aged ≥18 years (either hospitalized or receiving outpatient care) who were diagnosed with dyslipidemia and had no prior history of statin use. The mean age of participants was 61 years. Additionally, the patients had to meet inclusion criteria specifying LDL-C levels ≥ 100 mg/dL. In contrast, patients were excluded if they had a history of prior statin use, were currently taking statins, had undergone statin dose adjustments within the past 12 weeks, or lacked LDL-C measurement data. This helped ensure a statin-naïve cohort, reflecting truly untreated LDL-C levels that were not influenced by previous statin exposure.

### 2.2. Perspective, Time Horizon, and Discount Rate

The study adopted a 12-week treatment period based on the standard assessment timeline recommended in the 2024 Vietnamese Guidelines for the Management of Dyslipidemia issued by the Vietnam National Heart Association (VNHA) [19]. LDL-C assessment is recommended after at least 8 weeks to determine the next step, such as dose escalation or switching lipid-lowering medications. Given that hypercholesterolemia is a chronic condition, the time horizon was set to lifetime, in line with national recommendations and international guidelines (WHO, ISPOR), to capture the full costs and QALY gains associated with long-term statin use [20,21]. In this analysis, the lifetime horizon corresponds to 14 years, calculated as the difference between the mean age of the study cohort (61 years) [19] and the average life expectancy in Vietnam (75 years) [22]. Additionally, the research assessed cost-effectiveness from a healthcare perspective, using a 3% annual discount rate.

### 2.3. Model Structure

To simulate the treatment process and compare statins, a Markov model combined with a decision tree model was developed following the recommendation from the 2024 Vietnamese Guidelines for the Management of Dyslipidemia issued by the Vietnam National Heart Association (VNHA) [23], in combination with the Korean cholesterol management protocol [19]. Regarding treatment options, the model discussed the three interventions, corresponding with three statins: pitavastatin, atorvastatin, and rosuvastatin (as shown in Figure 1). Each statin was analyzed at two commonly used clinical doses: pitavastatin 2 mg and 4 mg, atorvastatin 10 mg and 20 mg, and rosuvastatin 5 mg and 10 mg.

Patients were initially treated with one of the three statins at starting doses: pitavastatin 2 mg, atorvastatin 10 mg, or rosuvastatin 5 mg for the first 8 weeks, according to VNHA guidelines [23]. The LDL-C goal attainment was assessed at the end of this period. Patients who failed to reach the LDL-C target were escalated to double the initial dose during the final 4 weeks, including pitavastatin 4 mg, atorvastatin 20 mg, or rosuvastatin 10 mg. The model then stratified patients into two outcomes: those who achieved or did not achieve the LDL-C target, followed by an assumption of differential cardiovascular event risk (stroke, myocardial infarction, or revascularization), and an estimation of corresponding quality-adjusted life years (QALYs). The model’s process, including the initial dose, assessment interval, and alternative treatment if LDL-C concentration is not controlled, is based on the 2024 Vietnamese Guidelines for the Management of Dyslipidemia by the VNHA [23].

### 2.4. Model Inputs

#### 2.4.1. Clinical Effectiveness

The clinical effectiveness of each treatment is calculated based on the probability of patients achieving the LDL-C target, which is summarized in Table 1. It is assumed that once patients successfully reduce LDL-C, the efficacy is maintained for the rest of the individual’s lifetime.

#### 2.4.2. Cost

The study employed direct medical costs only, including medication prices (based on the 2024 bidding report published by the Drug Administration of Vietnam [26]), and healthcare service costs (consultations and laboratory tests) as portrayed in Table 2, according to the Ministry of Health’s circular issued in 2023 [27]. All monetary values are presented in Vietnam Dong (VND), with the exchange rate in 2024 as 1 USD = 24,164.89 VND [28].

#### 2.4.3. Utility

The baseline utility weights in this study were stratified by age group (as shown in Appendix A) and adjusted based on major cardiovascular events, including stroke, myocardial infarction, and revascularization, which are potential complications of poorly controlled dyslipidemia (as demonstrated in Appendix A). No specific data were found to reflect changes in utility values following treatment with the pharmaceutical interventions under consideration. Therefore, the utility values for all treatment arms were assumed to be equivalent throughout the entire analysis period. To calculate the utility of each decision, we multiply the utility by the probability of that health state (achieving LDL-C target or not), with the formula as follows:Utility per branch = Utility Weight × Probability

For example, the utility of participants treated with pitavastatin 2 mg and achieving an LDL-C level was determined by the utility value in Appendix A and the corresponding probability (Table 1) as 0.8255 × 0.717 = 0.59188350. The comprehensive utility of each progress is portrayed in Table 3. The final utility of each intervention is determined by the total utility of that drug, which was further used to translate to the expected QALY (not displayed in Table 3). For example, the final utility of atorvastatin is the sum of atorvastatin 20 mg and atorvastatin 10 mg branches.

### 2.5. Statistical Analysis

Input parameters, including costs, treatment effectiveness, and utility values, were processed through expected cost and QALY calculations. In CUA, the primary outcome is the incremental cost-effectiveness ratio (ICER). ICER is a ratio of the cost required in exchange for a specific level of effectiveness, comparing two interventions. The ICER approach allows for comparing the economic advantage between two treatments that have distinct clinical efficacies. In this case, effectiveness is measured by QALY—a health outcome that ranges from 0 to 1, with 1 equivalent to perfect health, and any value less than 1 indicating a deteriorating health state [30]. The formula is presented as follows:$$\mathrm{ICERs}\;=\;\frac{Cost\;\left(A\right)\;-\;Cost\;\left(B\right)}{QALY\;\left(A\right)\;-\;QALY\;\left(B\right)}$$

Regarding cost, the total cost per intervention (which is called the expected total cost) is calculated based on the sum of costs multiplied by the probability of achieving effectiveness, as shown in the formula:Expected total cost per decision = cost × probability

As each intervention consisted of two doses, the total expected cost of pitavastatin would be measured based on the sum of the cost of pitavastatin 2 mg and pitavastatin 4 mg. A similar calculation would be applied in the atorvastatin and rosuvastatin cases.

On the other hand, the expected QALY of each intervention is calculated indirectly from utilities and the life years gained (LYG), with the formula:QALY = Utility × LYG

Specifically, LYG is determined by subtracting the mean age of the Vietnam population from the mean age of the study population. Following the 2024 Midterm Population and Housing Census Press Release by the General Statistics Office of Vietnam, the average age of the Vietnamese population is 75 years [22]. The mean age of the study population is 61 years.

Finally, the ICERs were then compared with Vietnam’s willingness-to-pay (WTP) threshold to determine the cost-effectiveness of each treatment strategy. The gross domestic product (GDP) of Vietnam measures the WTP. As the WHO recommends, the intervention that has an ICER surpassing ×3 times the GDP of the specific nation is deemed “not cost-effective”. The range within 1 to 3 times GDP is “cost-effective” and “very cost-effective” if the ICER falls below ×1 GDP. According to the financial report collected from the General Statistics Office (GSO) of Vietnam in 2024 [31]. Vietnam’s GDP per capita was VND 114,000,000; therefore, the WTP threshold used for the analysis in Vietnam was set at VND 342,000,000.

### 2.6. Sensitivity Analysis

A one-way sensitivity analysis was conducted using a Tornado chart to assess the model’s sensitivity to uncertainty in input variables. Each parameter of interest is changed while the other remain constant to evaluate the impact of that factor on the ICER range [32,33]. The variation is within ±20%. The results are portrayed in a Tornado chart, which helps researchers identify the most influential factor in the overall ICER value.

## 3. Results

### 3.1. Cost, QALY

The total expected cost for each intervention is summarized in Table 4, while a detailed breakdown of cost components per intervention is provided in Appendix A. The table presents a cost-effectiveness comparison of different statin interventions in terms of their total treatment cost and the probability of achieving target LDL-C levels. Pitavastatin 2 mg costs VND 69,203,400 (USD 2863.80) with a 71.7% success probability, while the 4 mg dose is more expensive at VND 94,582,600 (USD 3914.05) but improves the probability to 79.6%. Atorvastatin 10 mg and 20 mg have similar costs, around VND 81,677,000–81,786,000 (USD 3379.98–3384.50), with probabilities of 77.7% and 81.7%, respectively. Rosuvastatin shows lower costs, with the 5 mg dose at VND 46,095,980 (USD 1907.56) and 75.9% probability, while the 10 mg dose costs VND 74,381,230 (USD 3078.07) and achieves an 80.0% probability.

On the other hand, the calculated QALYs of each intervention are shown in Table 5. Although pitavastatin improved patients’ quality of life, its QALY was slightly lower compared to atorvastatin and rosuvastatin. All three interventions provide an equal survival benefit of 14 life-years gained. However, slight differences are observed in utility and QALY outcomes. Atorvastatin demonstrates the highest utility value (0.82070086) and QALY (11.28300399), followed closely by rosuvastatin with a utility of 0.81983168 and QALY of 11.19477268. Pitavastatin shows a slightly lower utility (0.81871072) and QALY (11.12292411).

### 3.2. ICERs

#### 3.2.1. Pitavastatin vs. Atorvastatin

The cost-effectiveness of pitavastatin compared to atorvastatin is presented in Table 6. Overall, using pitavastatin resulted in a cost saving of VND −31,280,140 over 14 years compared to atorvastatin. The incremental cost-effectiveness ratio (ICER) of pitavastatin was calculated to be 195,403,312 VND per QALY (8086.25 USD/QALY), which was within the ×1–3 GDP range of VND 114,000,000–342,000,000 (USD 4718.22–14,154.65). Consequently, pitavastatin is a cost-effective alternative to atorvastatin in the Vietnamese healthcare context.

#### 3.2.2. Pitavastatin vs. Rosuvastatin

The ICER result comparing pitavastatin and rosuvastatin is shown in Table 7. Similarly, the pitavastatin strategy required a substantially higher financial investment, yielding a lower QALY compared to rosuvastatin. The resulting ICER of −1,242,167,485 VND/QALY (−51,403.81 USD/QALY) indicates that pitavastatin was dominated by rosuvastatin, suggesting that rosuvastatin is both more effective and less costly in this comparison.

### 3.3. One-Way Sensitivity Analysis

The result of the one-way sensitivity analysis, illustrating the distribution of ICERs when adjusting input parameters for the comparison between pitavastatin and atorvastatin, is shown in Figure 2. The most influential factors were the drug costs of pitavastatin 2 mg and pitavastatin 4 mg, as well as utility. As the drug price of pitavastatin 2 mg was reduced, the cost-effectiveness improved significantly, with ICERs dropping by −184.85%. In contrast, laboratory test expenses were the least likely to alter the cost-effectiveness of the treatment.

Regarding the DSA result comparing pitavastatin and rosuvastatin in Figure 3, the utility as well as the drug cost of pitavastatin 2 mg and pitavastatin 4 mg were the most impactful parameters. The utility was particularly sensitive, as ICERs dropped substantially regardless of whether the utility increased or decreased. In contrast, medical costs had the least influence on the ICERs.

## 4. Discussion

### 4.1. Cost-Effectiveness

The study reveals that pitavastatin is more cost-effective than atorvastatin but more expensive than rosuvastatin over a lifetime horizon. Although pitavastatin yielded a slightly lower QALY, the difference was minor and offset by its lower cost compared to atorvastatin. In contrast, when compared to rosuvastatin, pitavastatin yielded a negative ICER (−1,242,167,485 VND/QALY), indicating that pitavastatin is both more effective and less costly, thereby establishing it as the dominant strategy in the current model. The divergence in cost-effectiveness trends when comparing pitavastatin to atorvastatin versus rosuvastatin stems primarily from differences in drug pricing, a factor confirmed by the one-way sensitivity analysis as having the most significant impact on ICER values. In Vietnam, rosuvastatin is widely available as a generic medication, while pitavastatin remains primarily confined to branded formulations. As a result, the price per tablet of pitavastatin 2–4 mg is at least 36% higher than that of rosuvastatin at a low dose of 5–10 mg.

Regarding the comparison with atorvastatin, the ICER of pitavastatin versus atorvastatin was within Vietnam’s willingness-to-pay threshold, suggesting pitavastatin would offer better economic benefit than atorvastatin. However, this conclusion holds only under the condition that the probability of achieving LDL-C targets remains high. In our study, a large proportion of patients successfully achieved LDL-C targets across all three treatment arms, with calculated rates of 94.22% for pitavastatin, 95.91% for atorvastatin, and 95.18% for rosuvastatin. These figures are consistent with previous estimates from a systematic review in Asia (which reported LDL-C goal attainment with pitavastatin ranging from 75% to 95%) [34]. The values reported in our model are at the upper end of that range, raising questions about real-world plausibility. Achieving a 95% success rate in LDL-C control would require patients to strictly adhere to the prescribed regimen throughout the entire 12-week treatment period and to sustain that adherence over the subsequent 14 years. In reality, however, non-adherence to statin therapy is common, with many patients discontinuing treatment or missing doses, which could compromise long-term effectiveness and, consequently, the reliability of the ICER projections.

Meta-analyses have revealed that adherence to statin therapy for cardiovascular disease prevention remains low, typically ranging between 50% and 60% [35]. Even for well-established statins like atorvastatin, which have been widely circulated in healthcare systems, adherence remains suboptimal, with real-world data showing a compliance rate of only 39%, and even lower (37%) among patients at high risk for cardiovascular disease [36]. Notably, a large-scale 2019 study in Taiwan assessing long-term statin use in over 180,000 patients following hospital discharge demonstrated significant declines in adherence over time. While 71% of patients were considered adherent in the first six months, this rate dropped to 51% at one year, and further declined to 42% by the end of year seven [37]. These findings underscore a substantial gap between the modelled assumption of sustained LDL-C target attainment and real-world treatment dynamics. As a result, the idealized ICER of pitavastatin, although favorable, may lack sufficient credibility to persuade healthcare providers or policymakers to shift from atorvastatin to pitavastatin, particularly when assessing long-term cost-effectiveness. Not to mention, Vietnam is classified as a lower-middle-income country [38], where the government is more likely to allocate its limited healthcare budget to address acute conditions with immediate mortality risks, rather than invest in managing risk factors with delayed or invisible outcomes, such as lipid control. In this context, pitavastatin is a cost-effective option; however, it does not appear to be affordable for implementation on a large scale in clinical practice.

However, the demonstrated cost-effectiveness of pitavastatin over atorvastatin may still offer value as an alternative treatment option. Our results estimated that patients treated with pitavastatin 2 mg had around a 71.7% chance of achieving the LDL-C target, which is similar to a Japanese study that reported a 60–70% success rate depending on cardiovascular risk levels [39]. As the Asian population is more sensitive to statins, lower doses already show strong LDL-C lowering; however, at the same time, patients are at a higher risk of musculoskeletal side effects [40]. Hence, clinicians are likely to initiate therapy at low to moderate doses to minimize muscle-related adverse effects [41,42]. One notable benefit of pitavastatin is its high tolerability, as it can avoid metabolism by CYP3A4, a common pathway for many other statins, including atorvastatin [43]. Metabolism via the CYP450 family often accelerates liver enzyme elevation and increases the risk of drug–drug interactions, making pitavastatin a promising and safer alternative to atorvastatin in older adults. Given the comparable LDL-C attainment rates between pitavastatin and atorvastatin, based on these findings, our analysis provides clinicians and healthcare specialists with a viable low-dose option that can achieve high LDL-C control rates for statin-naive individuals, particularly older adults or those at increased risk of intolerance.

Our conclusion about pitavastatin’s cost-effectiveness compared to the two most widely used statins was also observed in the Spanish cost-effectiveness analysis by Sales et al. in 2015 [44]. The authors found that rosuvastatin was the most economical option over both atorvastatin and pitavastatin in patients with high and very high ASCVD risk [44]. However, the context and the underlying decision question are distinct. Sales et al. applied a 25-year horizon and explicitly incorporated cardiovascular events into the model, which rendered pitavastatin non-cost-effective because it was less effective at preventing events and therefore failed to gain additional QALYs or reduce long-term CVD costs compared with rosuvastatin [44]. In contrast, our analysis did not model CVD events. Instead, we focused on LDL-C target attainment as the measure of effectiveness, and on drug acquisition and monitoring costs under Vietnamese prices. This framing shifts the interpretation: pitavastatin becomes less favorable than rosuvastatin because its higher price cannot be justified by comparable LDL-C lowering efficacy. This framework is particularly relevant for minority groups such as hypercholesterolemia in HIV patients, where clinicians must consider which statin has the fewest drug–drug interactions with antiretroviral therapy. In the REPRIEVE trial (2023), pitavastatin reduced the risk of major CVD events by 35% compared with the placebo in HIV patients [45]. Meanwhile, Yebyo et al. (2025) showed that other statins (rosuvastatin, atorvastatin, pravastatin, fluvastatin) provided about 21% CVD risk reduction in HIV patients but carried a 12% increased risk of type 2 diabetes [46]. On the other hand, in the cost-effectiveness study by Boettiger et al. [47,48]. Pitavastatin was not cost-effective versus a non-statin strategy in the US or Thailand (ICER exceeded WTP), but still dominated simvastatin in Thailand [47]. Taken together, this suggests that if the acquisition cost barrier can be reduced, pitavastatin could be a viable and cost-effective choice for achieving LDL-C goals in limited statin scenarios.

The added value of our study lies not only in providing evidence for pitavastatin as an economical and practical alternative for clinicians and patients, but also in supporting the Vietnamese healthcare system, particularly in hospital settings, in deciding which statins are most compatible with hospital budgets. In low- and middle-income country (LMIC) settings, where clinical efficacy and epidemiological data are often constrained, cost-effectiveness analyses become particularly valuable. The recent Indonesian study by Dewi et al. (2024) provided critical insights for LMIC health systems by demonstrating that, for ACS patients, the preventive effect of high-intensity statins on cardiovascular complications could outweigh the higher acquisition cost, justifying their inclusion in national health insurance schemes [49]. In Vietnam, however, the pathway for a drug to be reimbursed under the national health insurance depends on the hospital tendering process. According to Circular 15/2019/TT-BYT, drugs that win the tender are reimbursed for one year [50]. In this context, our finding that pitavastatin may be cost-effective compared with atorvastatin provides evidence to support its continued inclusion in future tenders, thereby offering an additional treatment option for patients who cannot use rosuvastatin or atorvastatin (such as those living with HIV). Given that pitavastatin most recently won a tender on 16 May 2024, our study provides policymakers with timely, updated information on the cost implications of maintaining pitavastatin as a reimbursed alternative.

### 4.2. Strengths and Limitations

The study employed a decision tree model to visualize the progression of health states, a strength when conducting a cost–utility analysis in dyslipidemia. Unlike Markov models, the decision tree assumes a one-way progression, where each health state occurs only once and does not repeat in cycles. This structure enables simplified assumptions and facilitates precise tracking of intervention effectiveness over a short-term horizon. In addition, the analysis compared pitavastatin with the two most widely used statins in clinical practice, providing clinicians with more insight into choosing the most effective statin for their patients. A further strength of our study is the focus on low-dose statin therapy rather than high-intensity regimens. This approach is consistent with prescribing trends in clinical practice in Vietnam. As a result, our findings are highly relevant to real-world treatment patterns and clinical expectations. Regarding cost, the drug prices were taken from Vietnam’s official databases, so the cost-effectiveness of pitavastatin calculated in this evaluation is relevant and applicable to the real-world healthcare context.

On the other hand, this study has several limitations that may affect the generalizability and practical applicability of its findings. First, our analysis did not incorporate cardiovascular events in the model, which we acknowledge as the main methodological drawback. As a result, the accumulated QALYs over a lifetime horizon may appear overly optimistic and do not fully reflect the actual health state of patients with hypercholesterolemia. The discrepancy between the dense evidence-based time-to-event data available for atorvastatin [51,52,53] and rosuvastatin [54,55] can explain this limitation in the general population, but the data are still very scarce for pitavastatin. So far, pitavastatin clinical evidence has been restricted to narrower populations, such as patients with stable coronary artery disease (CAD) [56] or those diagnosed with HIV [45]. In such a scenario, developing a comprehensive model that includes ASCVD events would rely on heavy extrapolation, making the results even less generalizable. Second, most of the input data were derived from international studies (e.g., LDL-C reduction effectiveness, utility weights, and complication probabilities). In contrast, local data from Vietnam remains limited, particularly in terms of cost profiles and patient quality of life. Differences in population characteristics, treatment conditions, and the healthcare system may result in model outcomes that do not accurately reflect real-world clinical practice in Vietnam. Additionally, this study focused solely on direct medical costs and did not account for indirect costs, such as productivity loss or other societal expenses. These limitations were incorporated to facilitate better integration of real-world Vietnamese data and comprehensive health economic analysis, thereby enhancing the reliability and applicability of the findings for pharmaceutical policymaking and clinical practice.

## 5. Conclusions

Pitavastatin is more cost-effective than atorvastatin but not more cost-effective than rosuvastatin over a lifetime horizon. Although choosing pitavastatin would result in lower QALYs, it reduces overall cost, with the calculated ICERs being over 195 million VND per QALY (nearly 8087 USD/QALY), substantially below Vietnam’s WTP. In the case of rosuvastatin, the higher cost of pitavastatin led to negative ICERS, indicating that rosuvastatin is the dominant option (lower cost, higher QALY). Drug cost is the main factor driving the ICERs. However, counting on ICERs alone would not be enough to evaluate the real economic effectiveness of pitavastatin, especially in the long-term and in LMIC settings.

## Figures and Tables

**Figure 1 healthcare-13-02494-f001:**
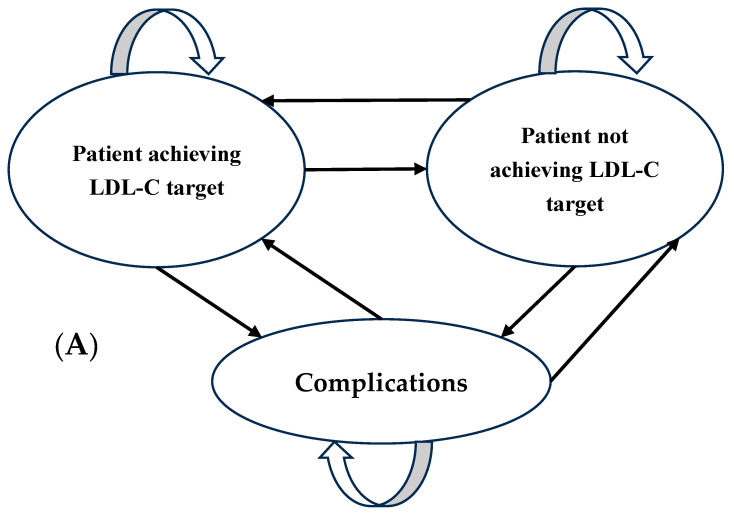
Markov model (**A**) and decision tree model (**B**) illustrating the health states of pitavastatin, atorvastatin, and rosuvastatin.

**Figure 2 healthcare-13-02494-f002:**
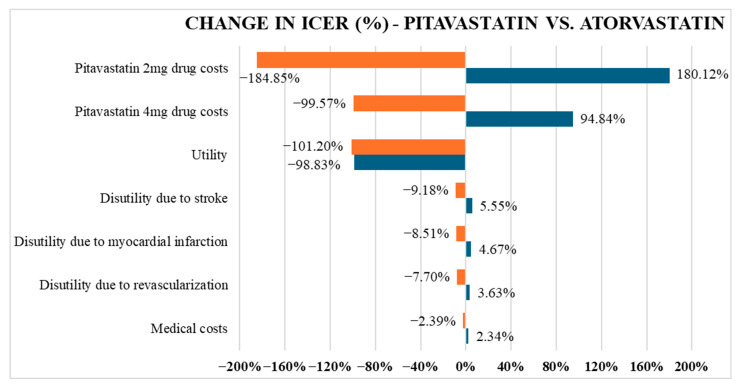
Tornado diagram comparing pitavastatin versus atorvastatin. When a variable value changes, the ICER result will change. The new ICER result will have a change in proportion to the original ICER value (the number on the graph). Orange indicates that the variable value has decreased by 20% from its original value, and blue indicates that the variable value has increased by 20% from its original value.

**Figure 3 healthcare-13-02494-f003:**
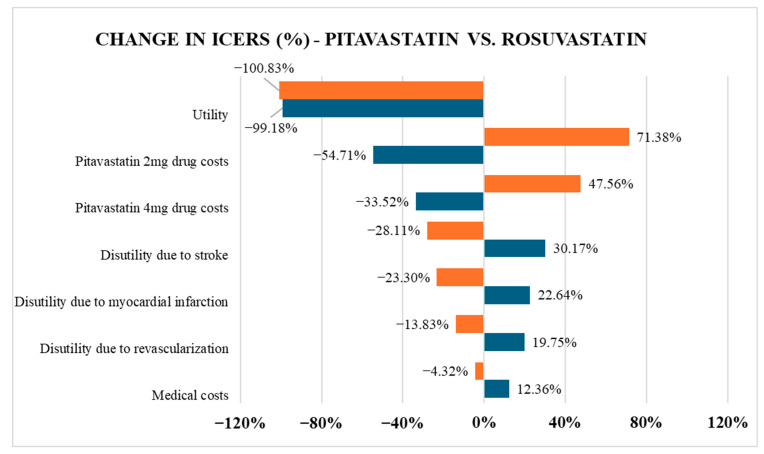
Tornado diagram comparing pitavastatin versus rosuvastatin. When a variable value changes, the ICER result will change. The new ICER result will have a change in proportion to the original ICER value (the number on the graph). Orange indicates that the variable value has decreased by 20% from its original value, and blue indicates that the variable value has increased by 20% from its original value.

**Table 1 healthcare-13-02494-t001:** Probability of controlling LDL-C level in the model.

Events	Probability *	*p* Value	Ref.
Patients achieving LDL-C target when treated with pitavastatin 2 mg	0.717	<0.001	[19]
Patients not achieving LDL-C target when treated with pitavastatin 2 mg	0.283	<0.001	Calculation
Patients achieving LDL-C target when treated with pitavastatin 4 mg	0.796	-	[24]
Patients not achieving LDL-C target when treated with pitavastatin 4 mg	0.204	-	Calculation
Patients achieving LDL-C target when treated with atorvastatin 10 mg	0.777	<0.001	[19]
Patients not achieving LDL-C target when treated with atorvastatin 10 mg	0.223	<0.001	Calculation
Patients achieving LDL-C target when treated with atorvastatin 20 mg	0.817	<0.001	[19]
Patients not achieving LDL-C target when treated with atorvastatin 20 mg	0.183	-	Calculation
Patients achieving LDL-C target when treated with rosuvastatin 5 mg	0.759	<0.001	[19]
Patients not achieving LDL-C target when treated with rosuvastatin 5 mg	0.241	<0.001	Calculation
Patients achieving LDL-C target when treated with rosuvastatin 10 mg	0.800	<0.001	[19]
Patients not achieving LDL-C target when treated with rosuvastatin 10 mg	0.200	-	Calculation
Patients with complications	0.145		[25]

*: Confidence of 95% CI.

**Table 2 healthcare-13-02494-t002:** Component of costs.

Type of Cost	Unit Cost in VND (USD)	Ref.
**Drug**		
Pitavastatin 2 mg	13,500 (0.56)	[26]
Pitavastatin 4 mg	18,500 (0.77)	[26]
Atorvastatin 10 mg	15,941 (0.66)	[29]
Atorvastatin 20 mg	15,941 (0.66)	[29]
Rosuvastatin 5 mg	8978 (0.37)	[29]
Rosuvastatin 10 mg	14,553 (0.60)	[29]
**Healthcare service**		
Total cholesterol test	27,300 (1.13)	[27]
HDL-C test	27,300 (1.13)	[27]
LDL-C test	27,300 (1.13)	[27]
Triglyceride test	27,300 (1.13)	[27]

**Table 3 healthcare-13-02494-t003:** Calculated utilities of each branch.

Branches	Utility	Ref.
Pitavastatin	0.81871072	Calculation
Atorvastatin	0.82070086	Calculation
Rosuvastatin	0.81983168	Calculation

**Table 4 healthcare-13-02494-t004:** Total cost of pitavastatin, atorvastatin, and rosuvastatin.

Intervention	Total Cost in VND (USD)	Probability of LDL-C Achieving the Target
Pitavastatin 2 mg	69,203,400 (2863.80)	0.717
Pitavastatin 4 mg	94,582,600 (3914.05)	0.796
Atorvastatin 10 mg	81,676,910 (3379.98)	0.777
Atorvastatin 20 mg	81,786,110 (3384.50)	0.817
Rosuvastatin 5 mg	46,095,980 (1907.56)	0.759
Rosuvastatin 10 mg	74,381,230 (3078.07)	0.800

**Table 5 healthcare-13-02494-t005:** Comparison of utility and QALY in 14-life-year-gained between pitavastatin, atorvastatin, and rosuvastatin.

Intervention	Utility	QALY
Pitavastatin	0.81871072	11.12292411
Atorvastatin	0.82070086	11.28300399
Rosuvastatin	0.81983168	11.19477268

**Table 6 healthcare-13-02494-t006:** Incremental cost, QALY, and ICERs between pitavastatin and atorvastatin.

Parameter	Pitavastatin	Atorvastatin
Total cost in VND (USD)	1,062,525,311 (43,969.80)	1,093,805,451 (45,264.24)
Total QALY	11.12292411	11.20300399
Incremental cost in VND (USD)	−31,280,140 (−1294.45)	
Incremental QALY (years)	−0.1600799	
**ICERs (VND/QALY)**	195,403,312	
**ICERs (USD/QALY)**	8086.25	

**Table 7 healthcare-13-02494-t007:** Incremental cost, QALY, and ICERs between pitavastatin and rosuvastatin.

Parameter	Pitavastatin	Rosuvastatin
Total cost in VND (USD)	1,062,525,311 (43,969.80)	973,277,353 (40,276.51)
Total QALY	11.12292411	11.19477268
Incremental cost in VND (USD)	89,247,958 (3693.29)	
Incremental QALY (years)	−0.0718486	
**ICERs (VND/QALY)**	−1,242,167,485	
**ICERs (USD/QALY)**	−51,403.81	

## Data Availability

No new data were created or analyzed in this study. Data sharing does not apply to this article.

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
