# Peer review of "Cost-Effectiveness Analysis of Pitavastatin in Dyslipidemia: Vietnam Case"

_healthcare, 2025, doi:10.3390/healthcare13192494_

Round 1
Reviewer 1 Report
Comments and Suggestions for Authors
All comments and revisions have been provided within the manuscript.
Nice study topic/methodology and scientific writing!

Author Response
Dear Reviewer 1, Please see the attachment. Thank you for your time to review our manuscript

Reviewer 2 Report
Comments and Suggestions for Authors
The manuscript presents a CUA of pitavastatin vs atorvastatin and rosuvastatin in the Vietnamese healthcare setting. The analysis is based on a decision tree model and 12-week time horizon. The time horizon is insufficient to capture long-term outcomes. All ISPOR and WHO guidelines emphasize that CUA should ideally adopt a lifetime horizon, or at least a horizon long enough to capture all the costs and outcomes. The chosen horizon is long enough to measure short-term LDL-C control costs but not cost-utility. The use of Korean utility data is also questionable. The data is more appropriate for a cost-effectiveness analysis or cost-minimisation (if there is data for equivalence), but definitely not for cost-utility analysis.
There are a lot of typos in the manuscript (for example, caculation instead of calculation). It is also not clear why authors discuss annual discounting rates with a 12-week time horizon. They also calculate the total QALY gain based on the lifetime horizon?
In my opinion, the method used in the analysis should be reconsidered.
Author Response
Dear Reviewer 2, Please see the attachment. Thank you for your time to review our manuscript

Reviewer 3 Report
Comments and Suggestions for Authors
THis manuscript addresses an important and good timed topic, evaluating the cost-utility of pitavastatin in Vietnam. Butseveral critical that I personally have a problem with, before this manuscript can see the light of the day:
Fisrt of all the time span is the most problematic thing for me, the 12-week horizon is too short for a cost-utility analysis of a chronic condition such as dyslipidemia. A longer time horizon (several years or lifetime) is standard practice in health economics, especially when outcomes such as cardiovascular events and quality of life are measured.
A lot of the data (utilities, clinical effectiveness) is derived from foreign studies rather than Vietnamese sources. I understand that the studies from Vietnam are scarce, if there are any more please make sure to involve them. If not, please reject this comment.
The model you presented here assumes long-term adherence and maintenance of LDL-C targets, while the study horizon is only 12 weeks. This inconsistency undermines the credibility of the analysis. Adherence rates should be modeled realistically, reflecting known challenges in long-term statin therapy. This goes alongside my first comment.
And one other thing, the conclusion that pitavastatin is cost-effective versus atorvastatin but dominated by rosuvastatin is already well established in the literature. The added value of this study is therefore limited unless the analysis is extended with a stronger local perspective and longer-term modeling. Again regarding my first comment.
English could use some more polishing.
Author Response
Dear Reviewer 3, Please see the attachment. Thank you for your time to review our manuscript

Round 2
Reviewer 2 Report
Comments and Suggestions for Authors
I have reviewed the revised manuscript and the authors’ cover letter. However, I find that the study contains critical methodological flaws. Decision trees are not appropriate for analyses conducted over a lifetime horizon, and as a result, the validity of the presented results is fundamentally compromised.
Author Response
Dear Reviewer 2,
Thank you for taking the time to review and encourage us to improve the manuscript. Please see the attachment.
Best regards,

Reviewer 3 Report
Comments and Suggestions for Authors
I am happy with the changes made.
Author Response
Dear Reviewer 3,
Thank you for taking the time to review and encourage us to improve the manuscript.
Best regards,